# Intelligent Machine Learning Approach for Effective Recognition of Diabetes in E-Healthcare Using Clinical Data

**DOI:** 10.3390/s20092649

**Published:** 2020-05-06

**Authors:** Amin Ul Haq, Jian Ping Li, Jalaluddin Khan, Muhammad Hammad Memon, Shah Nazir, Sultan Ahmad, Ghufran Ahmad Khan, Amjad Ali

**Affiliations:** 1School of Computer Science and Engineering, University of Electronic Science and Technology of China, Chengdu 611731, China; jpli2222@uestc.edu.cn (J.P.L.); or jalal4amu@yahoo.com (J.K.); muhammadhammadmemon@yahoo.com (M.H.M.); 2Department of Computer Science, University of Swabi, Swabi 23500, Pakistan; shahnazir@uoswabi.edu.pk; 3Department of Computer Science, College of Computer Engineering and Sciences, Prince Sattam Bin Abdulaziz University, P.O.Box. 151, Alkharj 11942, Saudi Arabia; s.alisher@psau.edu.sa; 4School of Information Science and Technology, Southwest Jiaotong University, Chengdu 611731, China; ghufraan.alig@my.swjtu.edu.cn; 5Department of Computer Science and Software Technology, University of Swat, Mingora 19130, Pakistan; amjad@uswat.edu.pk

**Keywords:** diabetes disease, feature selection, e-healthcare, decision tree, performance, machine learning, medical data

## Abstract

Significant attention has been paid to the accurate detection of diabetes. It is a big challenge for the research community to develop a diagnosis system to detect diabetes in a successful way in the e-healthcare environment. Machine learning techniques have an emerging role in healthcare services by delivering a system to analyze the medical data for diagnosis of diseases. The existing diagnosis systems have some drawbacks, such as high computation time, and low prediction accuracy. To handle these issues, we have proposed a diagnosis system using machine learning methods for the detection of diabetes. The proposed method has been tested on the diabetes data set which is a clinical dataset designed from patient’s clinical history. Further, model validation methods, such as hold out, K-fold, leave one subject out and performance evaluation metrics, includes accuracy, specificity, sensitivity, F1-score, receiver operating characteristic curve, and execution time have been used to check the validity of the proposed system. We have proposed a filter method based on the Decision Tree (Iterative Dichotomiser 3) algorithm for highly important feature selection. Two ensemble learning algorithms, Ada Boost and Random Forest, are also used for feature selection and we also compared the classifier performance with wrapper based feature selection algorithms. Classifier Decision Tree has been used for the classification of healthy and diabetic subjects. The experimental results show that the proposed feature selection algorithm selected features improve the classification performance of the predictive model and achieved optimal accuracy. Additionally, the proposed system performance is high compared to the previous state-of-the-art methods. High performance of the proposed method is due to the different combinations of selected features set and Plasma glucose concentrations, Diabetes pedigree function, and Blood mass index are more significantly important features in the dataset for prediction of diabetes. Furthermore, the experimental results statistical analysis demonstrated that the proposed method would effectively detect diabetes and can be deployed in an e-healthcare environment.

## 1. Introduction

Diabetes disease (DBD) is a significant health issue that many people suffer from around the world. The primary cause of this disease is associated with glucose level increase. One major cause of DBD (hyper-glycemia) is the deficiency of insulin, and beta cells in the pancreas producing insufficient insulin, which is called type-1 DB. In type-2 DB, the body cannot use the produced insulin accordingly [1]. DBD is the leading cause of different other critical complications, such as kidney disease, heart disease, neurological damages, damages to the retina and damage to feet and legs [2]. In 2014, about 422 million adults were suffering from DB, compared to 108 million in 1980. The diabetes disease increased from 4.7% to 8.5% in the adult population. DBD was the direct reason for the deaths of 1.6 million people in 2015, and in 2012, 2.2 million deaths were caused by high blood glucose [3]. In 2030, DBD will the 7th major cause of death [4]. The early detection of DBD is extremely important for effective treatments but all people with DBD are unaware of their condition until complications appear [5]. The complications of type-2 DBD can be prevented or delayed by detection at an early-stage and intervention in people at risk, see [1,5]. Thus, the early-stage detection of DBD is extremely important. To diagnose the DBD, various techniques have been adopted but all these techniques have some major drawbacks in detecting DBD in its initial stages. Thus, the intelligent analysis of medical data, including data-mining and machine learning methods are effective approaches for the detection of DBD. However, there are various factors to analyze for diagnosis of DBD and this complicates the job of physicians. The medical data and expert decision system to detect the DBD are the most important factors in the diagnosis of DBD. A review of the literature of the proposed diabetes techniques is good for understanding the significance of our suggested technique. All these prior recommended approaches used numerous methods to diagnose the diabetes. However, all of these approaches have a deficiency of prediction accuracy and require more execution time. The prediction accuracy of the diabetes identification technique needs further enhancement for efficient and accurate detection at early stages for better treatment and recovery. Thus, the key problems in these current methods are low accuracy and high computation time and these might be due to the use of non-suitable features in the dataset. To tackle these issues, new approaches are required to detect diabetes properly. The enhancement in prediction accuracy is a big challenge and a research gap. In this research study, we have designed an intelligent decision system based on machine-learning algorithms to successfully detect diabetes and to ensure a treatment in the early stages. Machine learning classifier DT has been used for classification. The Filter based DT (ID3) algorithm has been proposed for suitable features selection and its performances are high as compared to other feature selection techniques, such as DT ensemble Ada Boost [6], Random forest [7] and wrapper based feature selection method. Different validation methods, such as Hold out, K-Fold and Leave-One-Subject-Out (LOSO) have been used to select the best hyper parameters for the predictive model. Performance measuring metrics, such as classification Accuracy, Sensitivity, and Specificity, MCC, ROC-AUC, Precision, Recall, F1-score and Execution time are used to check the performance of the proposed system. The proposed system has been tested on the diabetes data set which is a clinical vital data set and designed from clinical obervations [8]. Additionally, the performances of the proposed method have been compared with the state of the art methods, such as LANFIS [9], TSHDE [10], C4.5 algorithms [11], Modified K-Means Clustering + SVM (10-FC) [12] and BN [13]. The experimental results demonstrated that the proposed method Filter based (DT-(ID3) +DT) achieved high classification accuracy compared with previous methods. All the experimental results are analyzed using statistical procedures.

The proposed research work is summarized in the following contributions/novelty:To propose a Filter based DT-(ID3) algorithm for features selection. The proposed algorithm should select more appropriate features from the dataset. Two ensemble algorithms, Ada Boost and Random Forest, are used for feature selection and compared the performance of DT on the proposed feature selection algorithm with these two FS algorithms and wrapper based feature selection methods.The Classification performance of the classifier has been checked according to original feature sets and on selected feature sets with cross validation methods, such as Hold out, K-fold, and LOSO. The LOSO is more suitable then train/test and k-folds validations. The classifier performance with the LOSO validation method is high in terms of accuracy of selected features compared to other validation methods such as Hold out and k-folds. Additional performance evaluation metrics results are very high with LOSO validation.

The remaining parts of the paper are organized as follows. Section 2 describes related work, Section 3 includes the proposed method to diagnosis diabetes, a brief explanation of the data preprocessing, the features selection algorithm, and the theoretical and mathematical background of machine learning classifiers. The validation procedures of classifiers, such as K-fold, LOSO, Hold out and statistical methods for comparing models are discussed in this section. The experimental setup and results are analyzed and discussed in Section 4. Finally, Section 5 shows the conclusion of the paper.

## 2. Related Work

Here, the related works for the diagnosis of DBD proposed by various researchers are briefly discussed. Kayaer and Yildırım [14] proposed a diabetes diagnosis system using different Artificial Neural Networks, Radial Basis Function and general regression neural network. The performance of GRNN was high compared to the Multi-layer perceptron (MLP) and RBF. The GRNN achieved 80.21% accuracy. Temurtas et al. [15] designed DBD, diagnosis system and used a Multilayer neural network structure by deploying the Levenberg-Marquardt (ML) algorithm and Probabilistic neural network architecture for classification of diabetes and healthy people. They used a 10-fold cross-validation method. Polat and Güneş [16] designed a two-stage diagnosis system and achieved 89.47% accuracy. In stage one, input features were reduced by applied principal component analysis algorithm and the second stage adaptive neuro-fuzzy inference system was deployed for DBD diagnosis. Sagir and Sathasivam [17] proposed an intelligent system for the diagnosis of diabetes using an adaptive network-based fuzzy inference system with Modified Levenberg Marquardt algorithm. The diagnosis system achieved 82.3% accuracy. Rohollah et al. [9] developed a Logistic Adaptive Network-Based Fuzzy Inference Diagnosis system applied samples with miss values and obtained 88.05% accuracy. Humar et al. [18] proposed a hybrid Neural Network System that was developed using Artificial Neural Network and Fuzzy Neural Network for diagnosis of DBD and obtained accuracy of 79.16%. Kemal et al. [19] developed a cascade learning system based on Generalization Discriminant Analysis (GDA) and Least Square Support Vectors machine (LS-SVM) for diabetes detection. Bankat et al. [11] designed a diagnosis system that used a K-mean Clustering algorithm to eliminate incorrectly classified samples from the data set. The C4.5 algorithm achieved a high accuracy of 92.38%. Yang et al. [20] developed a diagnosis system using the Bayes network and obtained 72.3% accuracy. Muhammad et al. [21] designed a three-stage system by using genetic programming with comparative partner selection for DB detection. A few methods have been proposed to generate a rule-based classification system. Wiphada et al. [22] designed a two stages rule generated system and this was confirmed on many UCI datasets. In the first step, neural networks nodes were pruned and analyses of the maximum weight and linguistic rules were created utilizing frequency interval data representation. The proposed method obtained 74% accuracy. Mostafa et al. [23] proposed a framework of learning rule from the dataset and achieved 79.48% accuracy. They designed the new update rule and focussed on the cooperation concept to generate strong rules. Fayssal et al. [24] developed a fuzzy classifier integrating with mutation operator to an Artificial Bee Colony algorithm for the creation of decision rule and obtained 84.21% accuracy. In [9], the authors developed sampling for the recursive rule extraction (Re-RX) integrated with the J48 graft algorithm for creation decision rules of the data set and achieved 83.83% accuracy. In [10] the authors proposed a two stage hybrid model of classification and decision rule extraction (TSHDE). They used a fuzzy ARTMAP classifier with Q learning known as QFAM in the first stage and used a genetic algorithm (GA) for rule extraction from QFAM in the second stage. The proposed method obtained 91.91% accuracy. Wei et al. [25] used the point process to treat the fMRI datasets of healthy controls and patients of diabetes, and then the functional brain network of subjects is designed using two sets of BLOD signals. The proposed method performances were good. Currently, optimization algorithms are using by researchers for decision rule generation. Binu et al. [18] developed an adaptive genetic fuzzy system (AGFS) for optimizing the rule and function of membership for the classification of medical data. Ramalingaswamy et al. [26] proposed a spider monkey optimization based rule miner (SM-RuleMiner) for diagnosis of diabetes and 89.87% accuracy was achieved. They have developed a novel fitness to calculate the fitness value of each candidate rule. Mohammad et al. [27], proposed hybrid method SVR using NSGA-II method for diabetes disease detection and achieved 86.13% accuracy. Ani et al. [28] designed IoT based E-healthcare system using ensemble classifier and the method attained 93% accuracy. Yang et al. [29] proposed an IoT cloud founded wearable ECG detecting method for smart e-healthcare. Khan et al. [30] proposed IoT based secure health care system to facilitate the best probable patient monitoring, efficient diagnosis, and timely diagnosis of patients.The controlling and treatment of diabetes disease [31] proposed MyDi framework which integrates a smart glycemic diary (for Android users), to automatically record and store patient activity via pictures and a deep-learning (DL)-based technology able to classify the activity performed by the patients via picture analysis. Similarly, in [32] developed an AI based method to interact with a patient (virtual doctor) by using a speech recognition and speech synthesis system and thus can autonomously interact with the patient, which is particularly important for, e.g., rural areas, where the availability of primary medical care is strongly limited by low population densities.

## 3. Materials and Method of Research

The following sub-sections contain the explanation of the materials and methods used in this paper. Mathematical notations used in the paper are summarized in Table 1.

### 3.1. Dataset

In this study, the diabetes dataset was used for modeling and testing the proposed method which is available on Kaggle machine learning repository [8]. Various preprocessing techniques have been applied before the feature selection process, such as min-max, variance, deviation, standardization, mean scaling and removal of missing values on the dataset [33,34].

### 3.2. Problem Statement of Feature Selection

The binary feature selection problem is described as follows: Let us consider diabetes disease dataset that have sample set $X=\{x_{1},x_{2},\ldots ,x_{n}\}$ and a finite set of *t* target label $Y=\{y_{0},y_{1}\}$ with *r* features $H=\{f_{1},f_{2},\ldots ,f_{r}\}$. The data set is expressed in Equation (Equation 1) as below:(1)$$F\left(X,Y\right)={\left\{\left(X_{i},Y_{i}\right)|X_{i}\in R^{n},Y_{i}\in \left\{y_{0},y_{1}\right\}\right\}}_{i=1}^{k}$$
where $X_{i}=\left\{x_{1},x_{2},\ldots ,x_{n}\right\}$$\in R^{n}$, are instances in the dataset and $Y_{i}\in {\left\{y_{0}=0,y_{1}=1\right\}}^{t}$ are output target classes labels in the dataset. In this equation, if $x_{i}$ has the target label $y_{j}$ then $y_{ij}=1$ otherwise $y_{ij}=0$. Additionally, $X={\left\{x_{1},x_{2},\ldots ,x_{n}\right\}}^{T}\in R^{n}$ is the instances matrix and $Y={\left\{y_{0},y_{1}\right\}}^{T}\in {\left\{0,1\right\}}^{n*1}$ is output label matrix. Figure 1 demonstrates the feature selection process.

#### 3.2.1. Proposed Filter Based Decision Tree Approach for Feature Selection

The relevant feature selection makes our approach more effective. The feature selection process is necessary for avoiding over fitting, to increase prediction performance and reduce the execution time of the classifier. Therefore, the major goal is to create a small subset $S=\left\{f_{1},f_{2},f_{3},\ldots ,f_{n}\right\}\left(p\leq r\right)$ containing enough representative information. To ensure that *S* can achieve optimal performance, it must possess Max-relevance and Minimum redundancy properties. The filter-based method measures the relevance of a feature by correlation with the dependent variable while the wrapper feature selection algorithm measures the usefulness of a subset of feature by actually training the classifier on it. The filter method is less computationally complex than the wrapper method. The feature set selected by filter is general and can be applied to any model and it is independent of a specific model. In feature selection, global relevance is of greater importance. To achieve these goals, we proposed a filter-based strategy using decision tree (DT) ID3 (Iterative Dichotomiser 3), Ada boost and Random forest algorithms for important features selection. The theoretical and mathematical background of these features selection algorithms is presented in the sections below.

##### Filter Based Decision Tree Iterative Dichotomiser 3 (DT-ID3) Feature Selection Algorithm

The ID3 algorithm begins with the actual data set *F* as the root node. In each iteration, it iterates through non used feature of the dataset *F* and computes the entropy H(F) or information gain IG(F) of that feature. Then ID3 selects the feature which has the smallest entropy or largest information gain value. The Set *F* is then divided by the selected feature to generate subset *S*. ID3 uses two metrics for measuring the feature importance, such as entropy and information gain [35,36]. The entropy (F) is a measure of the amount of uncertainty in the dataset *F* which expressed in Equation (Equation 2):(2)$$H\left(F\right)=\sum_{x\in X}-p\left(x\right)log_{2}p\left(x\right)$$
where *F* is the original data set for which entropy is being calculated, *X* is the features in the dataset *F*, and p(x) is the proportion of the number of elements in class *x* to the number of elements in the set *F*. When H(F)=0, the set *F* is perfectly classified. The information gain IG(F) is the measure of the difference in the entropy from before to after the Set *F* is split on feature *A*. It means how much uncertainty in set *F* was reduced after splitting set *F* on attribute *A*. Mathematically it is expressed in Equation (Equation 3).
(3)IG(F,A)=H(F)−∑t∈Tp(t)H(t)=H(F)−H(F|A)
where H(F) is entropy set *F*, *T* is the subsets generated from splitting set *F* by feature *A* such that $F=\cup_{t\in T}t$, P(t) is the proportion of the number of elements in t to the elements in *F*, and H(t) is the entropy of the subset *t*. The ID3 algorithm information gain can be computed for each remaining feature. The feature with high information gain is used to divide the set *F* on this iteration. We summarize the pseudo-code of feature selection for diabetes disease data set in Algorithm 1.
**Algorithm 1:** Filter Based DT-ID3 Approach for Feature Selection.
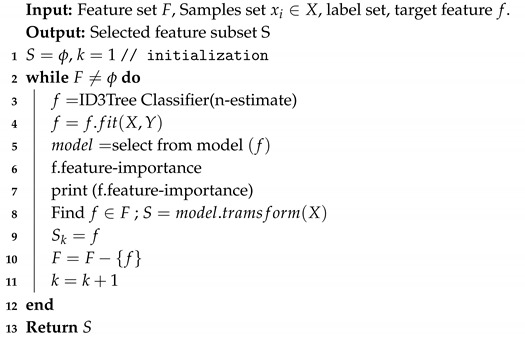


##### Ada Boost Feature Selection Algorithm

The Ada Boost (adaptive boosting) is ensemble decision tree algorithm [6]. It is also used for feature selection. The pseudo-Code of Ada Boost feature selection is given in Algorithm 2.

##### Random Forest Feature Selection Algorithm

Random Forests (RF) is an ensemble algorithm [7]. RF is also used for feature selection and the algorithm work as follows: at each node of the tree, it randomly selects some subsets of features f⊆F. where *f* is the set of features. The node divides the feature into subsets *f* instead of *F* and *f* is smaller than *F*. The procedures of features selection of RF features selection algorithm are given in Algorithm 3.
**Algorithm 2:** Ensemble Decision Tree Ada Boost FS algorithm.
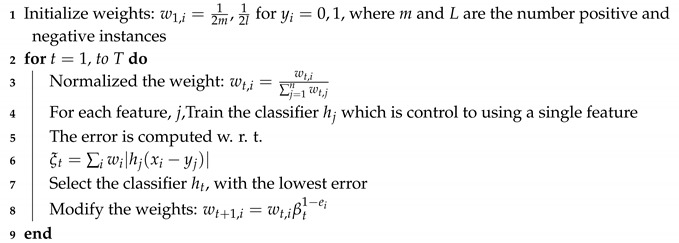


**Algorithm 3:** Ensemble Random Forest FS Algorithm.**1** Randomly select *f* features from *F* feature set where f⊆F****2**** The node *d* is computed using the best split point in features *f***3** Divide the nodes into sub nodes by using the best splits**4** Repeat the steps 1 to 3 until I number of nodes is reached**5** Create forest by repeating steps 1 to 4 for *n* number times to generate *N* number of trees.

#### 3.2.2. Wrapper Based Feature Selection Using Sequential Backward Selection Algorithm

Wrapper methods are based on greedy search algorithms as they evaluate all probable arrangements of the features. A wrapper-based sequential backward selection (SBS) is a standard feature selection algorithm, which comprehends the feature space into subspace feature with the lowest latency in classifier performance and reduces the model execution time. In some cases, SBS can increase the analytical ability of the model if a model faces an over-fitting problem [37]. SBS sequentially removes features from the full feature space until the new feature subspace has sufficient features. To determine which feature should be removed from feature space at each phase it is essential to define a function of criterion J to minimize. The criterion is calculated by the criterion that is simply being the variance in the performance of the classifier before and after the elimination of a specific feature. The feature that is removed at each phase can be defined as the feature that maximizes the criterion [38,39]. The pseudo-code of the SBS algorithm is given in Algorithm 4.
**Algorithm 4:** Wrapper based Sequential Backward Selection of Feature FS Algorithm.**1** Algorithm starting with k=d, the *d* is dimensional of feature full space $X_{d}$**2** Eliminate feature $x^{-}$, that maximizes the criterion:**3**$X^{-}=argmax\;\left(X_{k}-x\right)$, Where $x\in X_{k}$**4** Eliminate feature $x^{-}$ from feature space:**5**$X_{k}-1=X_{k}-X^{-}$**6**k=k−1**7** Finish if *k* reached the required features, if not then repeat step 2

### 3.3. Classification Algorithm

To classify diabetes and healthy people, we used the decision tree classifier in this study. A DT [40,41] is a supervised machine learning classifier, h:X→Y, which predicts the target labels related to sample *x* by traveling from root node of the tree to a leaf. A DT is mostly applied for classification problems [42,43,44,45,46,47]. DT is structured like a tree. For every node on the root to leaf path, the successor child is selected on the basis of a splitting of the input feature. Generally, the splitting is based on one of the features of x or the predefined set of dividing rules. The leaf node possesses specific information.

### 3.4. Cross Validation Methods

In this study, we applied three cross validation measuring methods, such as Hold out K-fold, and leave one subject out.

#### 3.4.1. Hold Out

In this validation method, the samples in the data set are split for training and testing of the classifier [48]. The 70% instances are used for training and 30% are used for validation of the classifier.

#### 3.4.2. K-Folds

In K- Folds [49] process data is split into K equal parts. The dataset split in K-1 and K-10 in each iteration for training and testing respectively. K times the process of validation executed. Average K calculation is performed to achieve the classifier performance. Here we use k = 10 in k Fold process. In the 10-Folds validation dataset, 90% is used for training and 10% for testing. Finally, at the end of the 10 folds’ process, the average value is calculated [50]. The average estimated performance is given and calculated through Equation (Equation 4).
(4)$$E=\frac{1}{10}\sum_{i=1}^{10}E_{i}$$

#### 3.4.3. Leave One Subject Out

LOSO is a cross validation special method in which Provides train/test indices to split data in train/test sets. Each sample is used once as a test set (singleton) while the remaining samples form the training set. This method is useful for the data set of small size.

### 3.5. Performance Evalution Matrix

To measure classification performance of the classifier, we use different metrics in this study, such as accuracy, specificity, sensitivity, Recall, precision, MCC, F1-score, ROC curve and processing time [42,48,49,51,52,53]. The binary confusion matrix has been used to computes these matrices.

The predicted output as True Positive (TP) when the diabetes subject is classified as diabetic, True Negative (TN) when the healthy subject is classified as healthy. False Positive (FP) if a healthy subject is considered a diabetes subject, similarly False Negative (FN) if the diabetes subject is considered a healthy subject. With the help of these four confusion matrices, performance evaluation matrices are computed.

**Accuracy (Acc):** Accuracy describes the overall performance of the classifier and mathematical accuracy expressed as below in Equation (Equation 5):(5)$$\mathrm{Acc}=\frac{\left(\mathrm{TN}+\mathrm{TP}\right)}{\left(\mathrm{TP}+\mathrm{TN}+\mathrm{FP}+\mathrm{FN}\right)}\times 100\%$$

**Sensitivity/Recall:** Sensitive show that the diagnostic test is positive and the person has diabetes disease and it also called True Positive Rate (TPR). Mathematically written in Equation (Equation 6): Sensitivity (Sn) /Recall/True Positive Rate (TPR):(6)$$\mathrm{Sn}=\frac{\mathrm{TP}}{\left(\mathrm{Tp}+\mathrm{FN}\right)}\times 100\%$$

**Specificity (Sp):** Specificity describes that a predictive test is negative and the person is healthy. The specificity and precision is expressed in Equations (Equation 7) and (Equation 8):(7)$$\mathrm{Sp}=\frac{\mathrm{TN}}{\left(\mathrm{TN}+\mathrm{FP}\right)}\times 100\%$$
(8)$$\mathrm{Precision}=p=\frac{\mathrm{TP}}{\left(\mathrm{TP}+\mathrm{FP}\right)}\times 100\%$$

**Major Complication or Comorbidity (MCC):** MCC shows the classifier predictability with value between [−1, +1]. If MCC is +1, it means the classifier predictions are ideal. If MCC is −1 which shows that classifier generates wrong predictions. If MCC is 0 it means that the classifier produces random predictions. The MCC is mathematically expressed in Equation (Equation 9):(9)$$\mathrm{MCC}=\frac{\left(\mathrm{TP}\times \mathrm{TN}-\mathrm{FP}\times \mathrm{FN}\right)}{\sqrt{\left(\mathrm{TP}+\mathrm{FP})(\mathrm{TP}+\mathrm{FN})(\mathrm{TN}+\mathrm{FP})(\mathrm{TN}+\mathrm{FN}\right)}}\times 100\%$$

**F1-score:** F1 score is the harmonic mean of precision and recall and mathematically expressed in Equation (Equation 10):(10)$$F1=\frac{2\mathrm{PR}}{\left(P+R\right)}$$
where *P* is representing precision while *R* is recall.

**ROC-AUC:** The ROC is a graphical tool for model performance analysis which compares the “True Positive Rate” and “False Positive Rate” in the classification results ML classifiers. AUC characterizes the ROC of the model. A high value of AUC shows a high performance of the model.

### 3.6. Methodology of the Proposed Technique for Diabetes Disease Detection

The major aim of the proposed research is to detect diabetes disease effectively. In the designing of the proposed technique Decision Tree algorithm has been used for suitable feature selection. The classifier Decision tree has been used for the classification of diabetes and healthy people. Cross validation methods, such as Hold out, K-fold and LOSO are used for the best hyper parameters tuning of the predictive model. Additionally, different evaluation metrics are used for model performance evaluation. The diabetes data set has been used for testing of the proposed method. Data preprocessing techniques are applied before feature selection. The overall procedures of the proposed method are given in Algorithm 5 and graphically shown in the flow chart in Figure 2. The following is the procedure for the proposed method for detecting diabetes and healthy people.
**Algorithm 5:** Proposed method for Diabetes detection.**1** Begin**2** Preprocessing of the Dataset using Different Statistical Techniques**3** Feature selection using DT (ID3) algorithm;**4** Using hold out, k folds and LOSO cross validation techniques for tuning hyper parameters and best model selection**5** Classification of diabetes and healthy people using DT classifier**6** Computes different performance evaluation metrics for model evaluation**7** Finish

## 4. Experiments and Results Discussion

The experimental setup and results are briefly discussed in the following sub-sections.

### 4.1. Experimental Setup

In this study, different experiments have been performed to identify diabetes disease. In these experiments, we performed data pre-processing using different statistical techniques. Then the processed dataset has been used for feature selection. The proposed ID3 algorithm has been used for feature selection. Classifier DT has been trained and tested on full and on selected feature sets to evaluate the performance of DT on full and on selected features. Different validation methods, such as hold out, K-fold and LOSO have been used for tuning hyper parameters and best model selection. Additionally, various model performance evaluation metrics have been computed automatically for model performance evaluation, such as accuracy, specificity, sensitivity, precision, recall, F1-score, MCC and ROC-AUC cure and processing time. The experimental results are tabulated and analyzed based on full and on selected feature sets. The result of the proposed method has been compared with the state of the art methods and different graphs were drawn for better presentation. Furthermore, different tools have been used for simulation of these experiments, such as Visio, Origin pro, and python on Intel @ R Core TM i5, 2400 CPU, 4 GB RAM with Window 10.

### 4.2. Experimental Results

All the experimental results are reported and discussed in the sub-sections below.

#### 4.2.1. Results of Pre-Processing Operations on the Dataset

The diabetes dataset has 2000 instances and 9 columns. The binary outcome column has two classes which take values ‘0’ or ‘1’ where ‘0’ for negative case means the absence of diabetes and ‘1’ for positive case means the presence of diabetes. The remaining 8 columns are real value attributes. Thus, the dataset is a 2000×8 features matrix. Furthermore, in the data set, 1316 are healthy subjects and 684 are diabetic subjects. The dataset was generated from Type 2 (DM1) diabetes patients. DM1 generally occurs in children but it can also appear in older people. In type 1 diabetes, subjects do not produce insulin and type 2 subjects do not have enough insulin.

The diabetes dataset instances and attributes along with some statistical information are described in Table 2. Furthermore, the visual representation of data set features are shown in Figure 3 and co-relation among the features of data set is visualized in Figure 4 using a heat map.

#### 4.2.2. Experimental Results of Feature Selection Algorithm Filter Based DT (ID3)

The proposed algorithm DT (ID3) has been used in order to select more appropriate features for correct and efficient classification of diabetic and healthy people. The proposed algorithm generates a subset of features and, on the selected features set, the classifier shows good performance compared to the whole features set. The proposed algorithm ranked all the features as shown in Table 3. Then the DT (ID3) algorithm selected important features from the whole features space. The selected features set contained features such as GL, AGE, IS, DPE, PG, BMI, and BP. The selected features of DT (ID3) are given in Table 4. These selected features are important for the detection of diabetes. The selected features are graphically shown in Figure 5 for better understanding.

#### 4.2.3. Experimental Results of Ensemble Ada Boost FS Algorithm

The Ada boost is an ensemble learning algorithm. It generates a small decision tree with a few features with the low computational process. The algorithm randomly selects some subset of the feature on the basis of feature weights. The features selected by Ensemble Ada boost are GL, BMI, DPF, IS, BP and AGE, i.e., five features. These features reported in Table 4.

#### 4.2.4. Experimental Results of Ensemble Random Forest FS Algorithm

The features selected by the Random Forest Algorithm are BP, GL, AGE, ST, IS, DPE, and BMI, which are important according to this algorithm. The features have been reported in Table 4.

#### 4.2.5. Experimental Result of Wrapper Based Sequential Backward Selection of Feature FS Algorithm

A wrapper-based algorithm discovers the feature space to score feature subsets according to their predictive power and optimizing the subsequent induction algorithm that uses the respective subset for classification. The feature subset selected by the wrapper based sequential backward selection algorithm are {GL,AGE,BMI,DPF,PG,IS}. According to this algorithm, these are important features for the diagnosis of diabetes. The feature ST and BP are not included in the selected feature sub set. Therefore, these features have a low impact in the diagnosis of diabetes.

#### 4.2.6. Classification Performance of Classifier DT with Individual Feature

In this section, the classifier DT performance has been checked with the individual feature in order to identify the individual importance of each feature of the data set in the prediction of diabetes. The individual prediction performance on each feature has been reported in Table 5. According to the table, the most prevalent features are DPF, GL, BMI, IS and AGE, and the classifier achieved high accuracy on these features. The Feature DPF achieved 84% test accuracy, 84% 10 folds average accuracy and 83% accuracy with the LOSO validation method. Similarly, the second most important feature is GL and the classifier DT achieved 75% accuracy only on this feature, 10 folds and LOSO based validation methods achieved 77% and 76% accuracy respectively. The third important feature in the dataset is BMI, and on this feature, the classification obtained 74% test accuracy, and 73% accuracy with k-folds where k is 10, and with the LOSO based method the achieved accuracy was 72%. Similarly, other important features in the data set are IS, AGE, PG, BP, and ST for which the classifier achieved good performance, respectively. Thus, according to classifier performance on individual features, we reached the conclusion that in this data set, DPF and GL are the most highly important features and these two features have great significance in the prediction of diabetes. The importance of these features is also indicated from Table 5 because the score values are high; the GL has a score of 0.23511 and DPF has a score value of 0.14366. The features such as GL, DPF, BMI have a low percentage of missing values and highly correlated features. The other features in the data set are of low importance and are loosely correlated to the target output variable. Further, these features have a low impact on the prediction of diabetes. The ROC curve and AUC values are high compared to other feature values. Thus, from Table 5, we concluded diabetesthat the feature GL and DPF are most important features in diabetes diagnosis and have great significant importance in the data set. If the features such GL and DPF are not considered in the prediction of diabetes then the predictive performance of DT will definitely be effected and give less accurate results. Additionally, according to Table 5 the feature selection algorithms also select these features for the effective detection of diabetes. However, the other features in the data along with these important features also have a great impact on the prediction performance of the classifier, DT for the diagnosis of diabetes. In Table 5, classification performance of the classifier, DT, has been checked for the full features set and feature set without GL. Thus, according to Table 5, the feature GL is critically important in the prediction of diabetes. The classifier achieved 97% test accuracy without GL and with GL it achieved 98.2%. A fasting blood sugar level less than 100 mg/L is normal. If fasting blood sugar level is between 100 and 125 mg/dL is considered normal and if its 126 mg/dL or higher the person has diabetes. Thus, the fasting blood sugar level value is used for the classification of diabetes and healthy people. Although in this work we used machine learning classifiers to classify diabetes and healthy subjects. The classifier prediction accuracy shows the overall performance of the system and the system accurately classifies healthy and diabetic subjects. The feature selection algorithm chooses suitable features for target classification. Therefore, the main aim of this work to classify the healthy and diabetic subjects using important features from the diabetes data set. The Feature GL, DPF, and BMI are selected by all feature selection algorithms. The feature ST according to Table 5 is a low significant feature in the prediction of diabetes.

#### 4.2.7. Classification Performance on Full Features Set and on Selected Features Sets Selected by Filter-Based Dt (Id3), Ada Boost And Random Forest

In these experiments, the DT classifier has been used for the classification of diabetes and healthy people. The performance of DT has been evaluated on the full and on the selected features set along with different cross-validation methods, such as hold out splits, k-folds and LOSO for best hyper-parameters tuning and for best model selection. In the train/test split method, 70% instances used for training and 30% instances were used for testing. Similarly, in k- fold the value of k = 10 was used. The model performance evaluation metrics have been computed and shown in Table 6. According to Table 6, the DT classifier on the full features set achieved 98.2% test accuracy while the selected features set selected by ID3 algorithm achieved 99% test accuracy. The specificity, sensitivity, and MCC on the full features set were 97%, 100%, and 99% respectively while on the selected features set these were 99%, 100%, and 99% which are high compared to the full features set. The precision, recall and F1-score results on the full features set were 99.8%, 100% and 100%. On the selected features set by (ID3) the values were precision 100%, recall 100% and F1-score 100% which is better than the full features set. The ROC-AUC value of DT on full features set was 99% while on selected features set (ID3) it was 99.8% which demonstrated that on selected features set the ROC-AUC value is good and covered more area than the ROC-AUC value on the full features set.

The 10-folds results of DT on full features set were 99.2% while on selected features set by (ID3) the 10-folds accuracy was 99.8% which is very good compared to the 10-folds value on the full features set. The LOSO validation accuracy on full features set was 99.6% while on the selected features set by (ID3) it was 99.9%, which demonstrated that the LOSO result is good for the selected features set compared to the LOSO results on the full features set. The execution time of DT on selected features set by (ID3) was 0.005 s while on the full features set the execution time was 0.006 s. Thus, the execution time of DT decreases on selected features. The classification accuracy of DT on the selected features set by FS ID3 with cross validation methods hold out, 10-folds, and LOSO are graphically shown in Figure 6 for better understanding which demonstrates that LOSO validation performance is good compared to the performances of hold out and K-fold validation. The LOSO validation achieved 100% accuracy. Another feature selection algorithm ADA BOOST selects important features of the data set which is reported in Table 4. The classifier performance has been checked on these selected features and reported in Table 6. The classifier DT achieved 98.5% test accuracy, 99.3% average accuracy of 10 folds and 99.6% accuracy with LOSO validation. Similarly, the feature selection algorithm RANOM FORET selected 7 important features from the data set, as we reported in Table 4. On this selected features set, the classifier performances have been checked and tabulated in Table 6.

According to experimental results on full features, the classifier DT with different validation, such as hold out, k-folds, and LOSO achieved 98.2%, 99.2% and 99.6% respectively, which is higher compared to the state of the art methods. Thus proposed DT classifier is more suitable for this dataset compared to other ML classifiers. Furthermore, the data preprocessing and feature selection mechanism improve the classification accuracy of DT with different validations, such as Training/testing, k-folds and LOSO achieved 99.0%, 99.8%, and 99.9% respectively. The improvement in classification accuracy is due to the selection of important features by the DT-ID3 FS algorithm. The ST feature according to DT-ID3 algorithm has a low impact in the prediction of diabetes. Thus, we think that the preprocessing and feature selection is critically important for significant improvement in the accuracy of the classifier. Due to the successful detection of diabetes by the proposed method (DT-ID3), we recommend the proposed method for efficient and accurate detection of DB in healthcare.

#### 4.2.8. Performance of Classifier on selected features set selected by Wrapper based Sequential Backward Selection algorithm

In this section, we embed the features selected by the wrapper-based SBS FS algorithm in classifier DT in order to check the performance of the classifier. The experimental results have been reported in Table 7. According to Table 7, the classifier DT achieved 98% test accuracy, 98.5% average accuracy with 10-folds and 98.9% accuracy with LOSO validation methods. Thus, we reach the conclusion on the basis of Table 6 and Table 7 that the performance of the Filter-based feature selection method with classifier DT is high compared to the Wrapper based feature selection method. Furthermore, the filter-based methods are computationally less complex compared to the wrapper methods and over fitting problems of filter based methods are low compared to the wrapper. Therefore, the proposed Filter-based DT-ID3 FS algorithm is more suitable for feature selection from the dataset because the number of features in the dataset is small.

#### 4.2.9. Performance Comparison of Our Method with Previous Methods for Diabetess Detection

The performance of the proposed method (DT (ID3)-DT) was compared with the existing methods in the literature in terms of accuracy for diabetes detection. The proposed method obtained good results in terms of accuracy. The accuracies of the proposed method with previous methods are given in Table 8. The proposed method achieved good performance in terms of accuracy and achieved 99% test accuracy, 99.8% k-folds average accuracy and 99.9% accuracy with LOSO validation. Hence, the proposed method could effectively diagnose diabetes. Furthermore, it can be easily incorporated into the smart health care system.

Statistically, to compare the performance of the proposed method with previously proposed methods in this study we used McNamara’s test [54,55]. Our hypothesis is that $H_{0}:n_{01}=n_{10}$, if the performance of DT(ID3-DT) and the other methods have the same accuracy.

In the alternate hypothesis $H_{1}:n_{01}\neq n_{10}$, the two models are very different. To test the null and alternate hypothesis we calculated the test statistic, or *p*-value. The value of alpha for all experiments is 0.05 and the confidence level 95%. Thus, on the basis of *p*-value and alpha, we accept or reject the null hypothesis on the following conditions

If p>α: then $H_{0}$ is fail to reject, the models have no difference.

If p≤α: then $H_{0}$ is rejected and alternate $H_{1}$ is accepted the models have different performance when trained on the particular training set R.

The test-statistic or *p*-value is calculated for each method and reported in Table 8. The significant level is 0.05. The DT-(ID3-DT) *p*-value is 0.04 and it is less than alpha. Other methods’ *p*-values are greater than the proposed method’s *p*-value. This means that the null hypothesis is rejected and the methods have significant differences in terms of accuracy. The smaller *p*-value of DT (ID3-DT) than alpha demonstrated that DT-(ID3-DT) is more significant than previous approaches

## 5. Conclusions

Machine learning data mining techniques play an important role in healthcare services by delivering a system to analyze the medical data for diagnosis of diseases. The successful detection of diabetes is a critical medical issue for medical experts and researchers. To tackle this problem, we have proposed an E-healthcare system for the detection of diabetes using ML data mining techniques. In the proposed method, we have used the DT (ID3) algorithm for features selection as features selection is necessary for effective training and testing of the classifier. Additionally, ensemble learning DT Feature selection algorithms Ada Boost and Random Forest are also used for feature selection. The DT machine learning classifier has been used for the detection of diabetes. The DT has no need for extra parameters during the training and testing process. Additionally, we used different cross-validation techniques to validate the predictive model, such as hold out, K-fold, and LOSO. To check the model classification performances, various performance evaluation metrics have been used in this study, such as accuracy, specificity, sensitivity, MCC, ROC-AUC, precision, recall, F1-score and execution time. The diabetes dataset was used to check the proposed method. The experimental results analysis demonstrated that the proposed feature selection algorithm Filter Based DT (ID3) selects more suitable features and the classifier DT achieved good performances on these selected features as compared to feature sets selected by Ada Boost and Random Forest algorithms. The Features GL, DPF and BMI are more significantly important features in dataset and have great influence in the detection of diabetes and all features selection algorithms select these features. The feature ST has an impact in the detection of diabetes and two FS algorithms did not select it. The proposed method DT (ID3) +DT achieved 99% test accuracy, 99.8% accuracy with k-floods and 99.9% accuracy with LOSO validation. Furthermore, the classifier DT performance with Filter-based feature selection method is high compared to the wrapper-based feature selection method in terms of accuracy and computation time. The experimental results of metrics used in this research are good enough. Statistical analysis showed that the performance of the proposed method in terms of accuracy is good compared to the previously proposed methods. Thus, the results of the proposed research suggest that the proposed method is more suitable for the detection of diabetes in healthcare. In the future, we will use an embedded based feature selection method in order to select an important feature from the data set. The proposed method will also be applied for other data sets, such as Parkinson’s, heart disease, and breast cancer for efficient and accurate diagnosis of these diseases. Additionally, after the diagnosis of disease, proper treatment is extremely import for better recovery. In future work, we will design treatment and recovery methods for critical diseases.

## Figures and Tables

**Figure 1 sensors-20-02649-f001:**
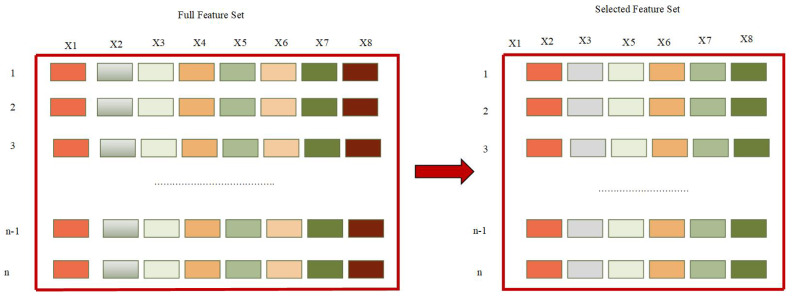
Feature selection process.

**Figure 2 sensors-20-02649-f002:**
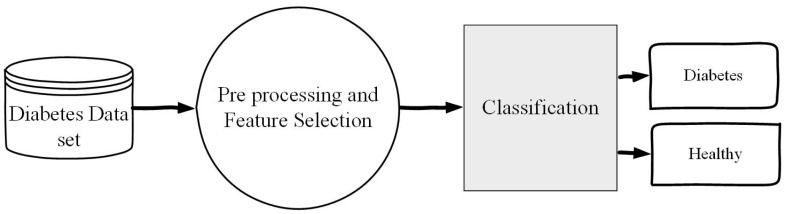
Flow chart of the proposed method of Diabetes Detection.

**Figure 3 sensors-20-02649-f003:**
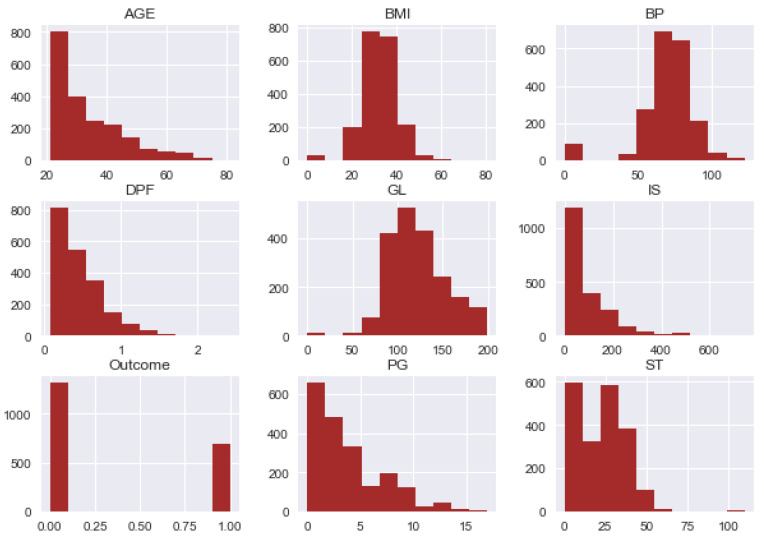
Histograms for the visual representation of features.

**Figure 4 sensors-20-02649-f004:**
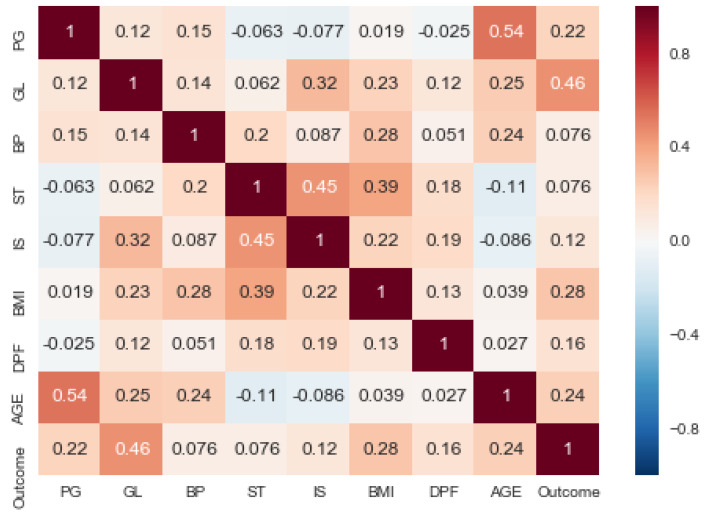
Heat map of the dataset.

**Figure 5 sensors-20-02649-f005:**
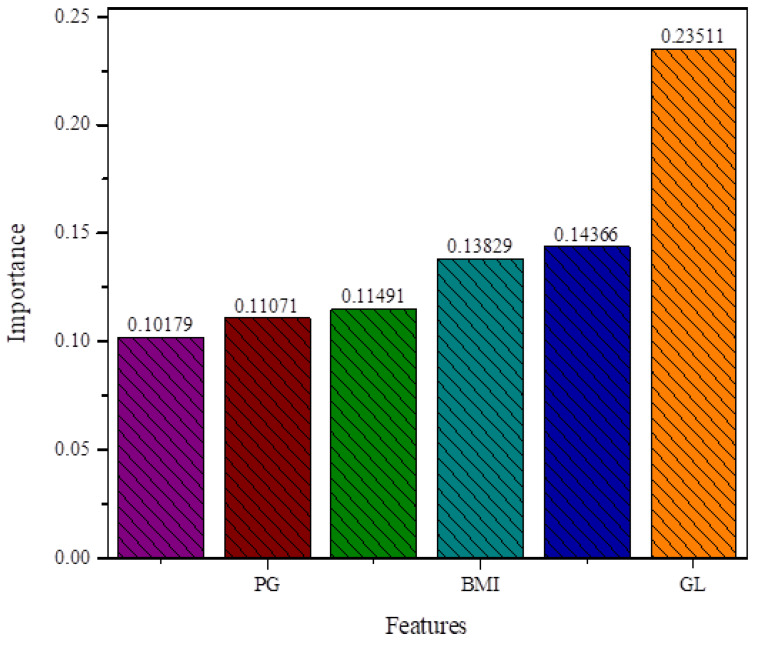
Feature selected by DT (ID3) algorithm.

**Figure 6 sensors-20-02649-f006:**
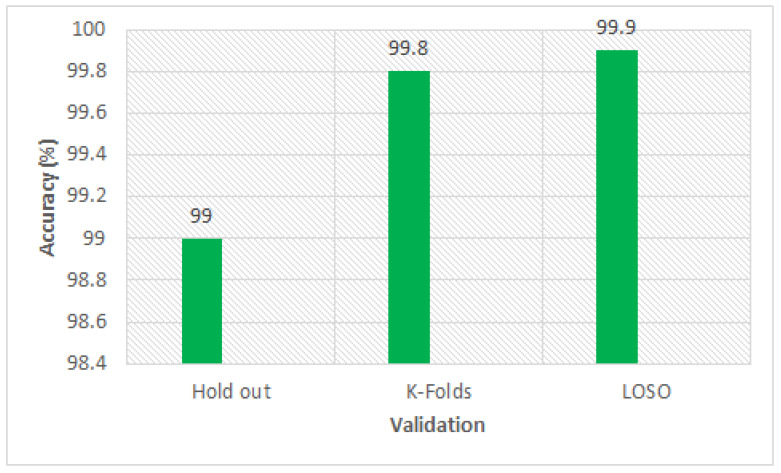
Accuracy on selected features set by DT-ID3 with different validation methods.

**Table 1 sensors-20-02649-t001:** Mathematically symbols and notations Used in the paper.

Symbol	Description
*H*	Data set
*S*	Subset
*F*	Feature set
*n*	Number of instances in dataset
*X*	Input features in dataset
*Y*	Predicted output classes label
*b*	Bais is offset value from the origin
*w*	d-dimensional coefficient vector
*i*	*i* is *i*th sample in data set
$x_{i}$	*i*th instance of dataset sample X
$y_{i}$	Target labels to x
*R*	Training set
*T*	Test set
*t*	Finite set
IG(F)	Information gain
*p*-value	Test probability value
α	Degree of freedom
*f*	Feature in dataset
MI	Mutual information
$F_{i}$	ith feature in dataset
ϕ	Empty sect
*p*	probability
$H_{0}$	Null hypothesis
$H_{1}$	Alternate hypothesis

**Table 2 sensors-20-02649-t002:** The Diabetes dataset description along with some statistical operations.

Feature Name	Feature Code	Description	Min-Max	Mean, (±) STD
Pregnancies	PG	Number of period pregnant	0.000000–17.000000	3.703500, (±) 3.306063
Glucose	GL	Plasma glucose concentrations	0.000000–199.000000	121.182500, (±) 32.068636
Blood Pressure	BP	Blood pressures (mm Hg)	0.000000–122.000000	69.145500, (±)19.188315
Skin Thickness	ST	Triceps skin fold thickness(mm)	0.000000–110.000000	20.935000, (±) 16.103243
Insulin	IS	Serum insulin concentration	0.000000–744.000000	80.254000, (±)111.180534
BMI	BMI	Blood mass index	0.000000–80.600000	32.193000, (±) 8.149901
Diabetes Pedigree Function	DPF	Diabetes pedigree function	0.078000–2.420000	0.470930, (±) 0.323553
Age	AGE	Age in years	21.000000–81.000000	33.090500, (±)11.786423
Outcome	1 = yes	Diabetes = 1	0.000000–1.000000	0.342000, (±) 0.474498
	0 = no	Healthy = 0		

**Table 3 sensors-20-02649-t003:** Feature ranking and importance by decision tree (DT) (ID3) algorithm.

S.No	Feature Label	Ranking	Score
1	PG	IS	0.07605
2	GL	ST	0.07947
3	BP	BP	0.10179
4	ST	PG	0.11071
5	IS	DPF	0.11491
6	BMI	BMI	0.13829
7	DPF	AGE	0.14366
8	AGE	GL	0.23511

**Table 4 sensors-20-02649-t004:** Rank and score of features selected by DT (ID3), Ada Boost and Random Forest algorithm.

S.NO	Feature Set	Feature Selection Algorithm		
		DT(ID3)	Ada Boost	Random Forest
1	PG	GL	GL	BP
2	GL	AGE	BMI	GL
3	BP	IS	DPF	AGE
4	ST	DPF	BP	ST
5	IS	BMI	AGE	IS
6	BMI	BP	IS	BMI
7	DPF	PG		DPE
8	AGE			

**Table 5 sensors-20-02649-t005:** Classification Performance on individual features, full features and features set without GL.

Classifier	Feature	Acc (%)	Sn (%)	Sp (%)	MCC (%)	ROC-AUC (%)	K-Fold (%)	LOSO (%)	Time (s)
DT	GL	75	45	88	67	67	77	76	0.001
	BP	68	8	74	52	53	67	66	0.005
	BMI	74	45	88	66	66	73	72	0.005
	DPF	84	66	87	78	78	84	83	0.002
	IS	73	34	92	64	63	73	73	0.001
	ST	68	14	95	54	54	65	66	0.001
	PG	69	27	90	59	58	69	70	0.0009
	AGE	70	40	85	62	63	70	71	0.0018
	Full with GL	98.2	100	97	99	99	99	99.8	0.006
	Without GL	97	75	82	97	97	99.5	99.7	0.005

**Table 6 sensors-20-02649-t006:** Classification Performance with and without selected feature set by Filter FS algorithms.

Feature Set Selection	Acc (%)	Sn (%)	Sp (%)	MCC (%)	Pre (%)	Rec (%)	F1 (%)	ROC (%)	K-Folds (%)	LOSO (%)	Time (S)
Full set	98.2	98	97	97	99.8	98	98.6	98	99.2	99.6	0.006
ID3	99	100	98	99	100	100	100	99.8	99.8	99.9	0.005
Ada Boost	98.5	98	99	98	98	98	99	98.6	99.3	99.6	0.004
Random Forest	98.3	98	98	98	95	98	99	98.7	99.4	99.7	0.006

**Table 7 sensors-20-02649-t007:** Classification Performance with and without selected feature set by Wrapper based FS algorithms.

Feature Set Selection	Acc (%)	Sn (%)	Sp (%)	MCC (%)	Pre (%)	F1 (%)	ROC (%)	K-Fold (%)	LOSO (%)	Time (s)
SBS	98	99	98	98	99	98	97.6	98.5	98.9	0.007

**Table 8 sensors-20-02649-t008:** Performances comparison of the proposed method with previous methods on the diabetes dataset.

Reference	Method	Accuracy (%)	*p*-Value
[9]	LANFIS	88.05	0.87
[26]	SM-Rule-Miner	89.87	0.92
[10]	TSHDE	91.91	0.21
[11]	C4.5 algorithm	92.38	0.69
[12]	Modified K-Means Clustering +SVM (10-FC)	96.71	0.07
[56]	Support Vector Machine	97.14	0.06
[57]	Artificial Neural Network (ANN)	82.35	1.23
[58]	SBNN + PSO + ALR	88.75	0.31
[59]	DPM	96.74	0.08
[60]	DNN	95.6	0.09
[13]	BN	99.51	0.06
	DT(ID3) + DT	99 (Hold out)	0.04
Our study	DT(ID3) + DT	99.8 (K-fold)	
	DT(ID3) + DT	99.9 (LOSO)

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
