# Peer review of "Intelligent Machine Learning Approach for Effective Recognition of Diabetes in E-Healthcare Using Clinical Data"

_sensors, 2020, doi:10.3390/s20092649_

Round 1

Reviewer 1 Report

Comment #4 has been partially addressed. A separate section has been created for related work. Apart from that, my initial concerns have not been satisfactorily addressed.

Figures and tables are still not properly formatted. In most cases figures are not properly separated from text above and below them.  For example, no separation between Table 10’s title and text above it!

 The figure below Table 8 has very poor resolution. In addition, the figure appears not to  have been titled as its title is in the following page!

 Readability of the article has not been significantly improved.

 What do the authors mean by “Machine learning data mining”?

 Equation on page 9 should be properly formatted.

 Equation 12 is wrong, what is r?

Author Response

Reviewer #1 comments:

Comment # 1:

comment #4 has been partially addressed. A separate section has been created for related work. Apart from that, my initial concerns have not been satisfactorily addressed.

Reply: Thank you.  According to previous review, the title and abstract of the manuscript have been updated and mentioned the reasons. Further, about the novelty of the proposed work, this work has the following contributions.

  1. To design ML based e-health care diagnosis method for diabetes diagnosis.
  2. To propose Filter based DT-(ID3) algorithm for features selection and the proposed algorithm select more appropriate features from the dataset. Also, two ensembles algorithms, such as Ada Boost and Random Forest are used for feature selection and compared the performance of DT on the proposed feature selection algorithm with these two FS algorithms and wrapper based feature selection method.
  3. To use the classifier DT and the performance have been checked on original features set and on selected features set along with cross validation methods, such as Hold out, K-fold, and LOSO. The LOSO is more suitable then Hold out and k-folds validations. The classifier performance with LOSO validation method in terms of accuracy on selected features is high as compared to other validation methods, such as train/test and k-folds. Additional other performance evaluation metrics results are very high with LOSO validation.

Comment # 2:

Figures and tables are still not properly formatted. In most cases figures are not properly separated from text above and below them.  For example, no separation between Table 10’s title and text above it!

Reply: Thank you. All the figures and tables adjusted in updated manuscript accordingly.

Comment # 3:

The figure below Table 8 has very poor resolution. In addition, the figure appears not to have been titled as its title is in the following page!

Reply: Thank you. The figure 8 adjusted redrawn professionally in updated manuscript.

Comment # 4:

Readability of the article has not been significantly improved.

Reply: Thank you. The revised paper is formatted in latex format of the journal and readability is improved.

Comment # 5:

What do the authors mean by “Machine learning data mining”?

Reply: Thank you. Rectified it in updated manuscript as Machine learning data mining techniques played important role in healthcare services which delivers a system to analyze the medical data for diagnosis of diseases

Comment # 6:

Equation on page 9 should be properly formatted.

Reply: Thank you. Equation 9 is correctly formatted in updated manuscript.

Comment # 6: Equation 12 is wrong, what is r?

Reply: Thank you. The equation 12 is corrected accordingly in update manuscript.

R is recall.             

Reviewer 2 Report

I would like to thank the authors for revising the manuscript. However, some of my concerns were not addressed.

Please, add you answer after EACH of my comments. Otherwise, it is really difficult to understand if you replied appropriately. 

English is really poor. To me, the paper cannot be published as it is. Proofreading is mandatory.

Title

I don’t agree with using “new” in the title. The title should contain information on the processed data and the machine learning methods.

Abstract

In the abstract, you have to write if you are dealing with EHRs, data from sensors, images... "diabetes data set" is not meaningful.

The acronyms in the abstract are still undefined. This is really disappointing.

Introduction

In the introduction (line 55): still, at this point, it is not clear if you are dealing with type 1 or type 2 diabetes.

Line 62-70: This paragraph makes sense ONLY after the literature review (as it is now, this is just an authors' opinion). My comment is still valid: Considering what the authors propose, this is not a strong discussion on the limits in the literature. For example, the prediction accuracy values of methods in the literature highly depend on the processed dataset: you cannot compare performance obtained with different datasets. Similarly, the computational time depends on the hardware. The authors are proposing a machine-learning algorithm exactly as other researchers do in the literature. So, the innovation with respect to the literature is not clear. 

Line 71: The data you are using should be indicated here (which data? EHRs, data from sensors, images?)

Sec. 3.4

As already stated in my previous review, you don't need a paragraph to describe these approaches. There are well known from researchers working in ML. 

I don't agree with your sentence: "We used these three cross validation methods, in order to check which validation method is suitable for this data set. ". What do you mean by checking the method that is more suitable? The reason why one should use cross validation is to build a statistics on your data. There are not “more or less suitable methods”. I strongly suggest the authors to find a better justification of using these 3 approaches. Otherwise, they should just present one approach.

Figure 2 does not make any sense here. If you want to keep it, you should improve it (it is not that informative now, e.g., you can add section number...)

Author Response

Reviewer # 2 comments:

I would like to thank the authors for revising the manuscript. However, some of my concerns were not addressed.

Please, add you answer after EACH of my comments. Otherwise, it is really difficult to understand if you replied appropriately. 

English is really poor. To me, the paper cannot be published as it is. Proofreading is mandatory.

Comment # 1:

Title

I don’t agree with using “new” in the title. The title should contain information on the processed data and the machine learning methods.

Reply: Thank you.  The title is modified accordingly in updated manuscript.

Intelligent Machine Learning Approach for Effective Recognition of Diabetes in the E-HealthCare Using Clinical Data”

Comment # 2:

Abstract

  1. In the abstract, you have to write if you are dealing with EHRs, data from sensors, images... "diabetes data set" is not meaningful.

     Reply: Thank you. The dataset has been generated from clinical observations and it’s a vital data set and available of kaggle ML repository. The detail data of data set available in section 4.2.1

  1. The acronyms in the abstract are still undefined. This is really disappointing.

Reply: All acronyms defined in abstract accordingly.

Abstract: A significant attention has been made to the accurate detection of diabetes which is a big challenge for the research community to develop a diagnosis system to detect diabetes in a successfully way in the E-healthcare environment. Machine learning data mining techniques have an emerging role in healthcare services which delivers a system to analyze the medical data for diagnosis of diseases. The existing diagnosis systems have some drawbacks, such as high computation time, and low prediction accuracy. To handle these issues, we have proposed a diagnosis system using machine learning methods for the detection of diabetes disease. The proposed method has been tested on the diabetes data set which is a clinical dataset designed from patient’s clinical history. Further, Model validation, such as hold out, K-fold, leave one subject out and performance evaluation metrics, includes accuracy, specificity, sensitivity, F1-score, receiver operating characteristic curve, and execution time have been used to check the validity of the proposed system. We have proposed a filter method based on the Decision Tree (Iterative Dichotomiser 3) algorithm for highly important feature selection. Two ensemble learning algorithms, such as Ada Boost and Random Forest are also used for feature selection and compared the classifier performance with wrapper based feature selection algorithms also. Classifier Decision Tree has been used for the classification of healthy and diabetic subjects. The experimental results show that the proposed feature selection algorithm selected features improve the classification performance of the predictive model and achieved optimal accuracy. Additionally, the proposed system performance is high as compared to the previous state-of-the-art methods. High performance of the proposed method is due to the different combinations of selected features set and Plasma glucose concentrations, Diabetes pedigree function, and Blood mass index are more significantly important features in the dataset for prediction of diabetes disease. Furthermore, the experimental results statistical analysis demonstrated that the proposed method would be effectively detected diabetes disease and can easily be deployed in E-healthcare environment.

Comment # 3:

Introduction

  1. In the introduction (line 55): still, at this point, it is not clear if you are dealing with type 1 or type 2 diabetes.

    Reply: Thank you. The data set is clinical data set type and the diabetes type 2 patients.

  1. Line 62-70: This paragraph makes sense ONLY after the literature review (as it is now, this is just an authors' opinion). My comment is still valid: Considering what the authors propose, this is not a strong discussion on the limits in the literature. For example, the prediction accuracy values of methods in the literature highly depend on the processed dataset: you cannot compare performance obtained with different datasets. Similarly, the computational time depends on the hardware. The authors are proposing a machine-learning algorithm exactly as other researchers do in the literature. So, the innovation with respect to the literature is not clear.

Reply: Thank you. The performance of proposed method compared with existing methods in term of accuracy and some of these methods used the same data set and some used pima Indian diabetes dataset. Both data sets have same number of features, however, numbers of instance the data set used in this work is more than pima Indian diabetes data set. According to literature different ML feature section and classification techniques used for detection of diabetes. The proposed method achieved high performance due the selection of appropriate features by proposed FS algorithm. Secondly the proposed method achieved high accuracy with LOSO validation method and reason of high accuracy is that LOSO is more suitable for small data set. The authors methods used others validation techniques and due these validation techniques exiting methods performance are low as compared to our method.

Further, the computational time of a method defend on the designing of the method and hardware’s system use for simulation. According our experimental setup, the computation time our method is very low as compared to other method.

Due to the high prediction accuracy of the proposed method we recommend it for accurate and efficient detection of diabtesats disease. The improvement in accuracy we think that is great contribution of the proposed method.

  1. Line 71: The data you are using should be indicated here (which data? EHRs, data from sensors, images?)

Reply: Thank you. the diabetes data set is clinical vital data set and designed from clinical observations

For clinical examination of 2000 people various methods were used, such as CT Scans, MRIs, X-rays, and other trails methods, on the basis of the image data the data set is deigned in which 1316 are health people and 684 people have diabetes.

Comment # 4:

. Sec. 3.4

  1. As already stated in my previous review, you don't need a paragraph to describe these approaches. There are well known from researchers working in ML. 

Reply: Thank you. We incorporated your suggestion.

  1. I don't agree with your sentence: "We used these three cross validation methods, in order to check which validation method is suitable for this data set. ". What do you mean by checking the method that is more suitable? The reason why one should use cross validation is to build a statistic on your data. There are not “more or less suitable methods”. I strongly suggest the authors to find a better justification of using these 3 approaches. Otherwise, they should just present one approach.

Reply: Thank you.  The three validation techniques we used for the purpose that which one give high performance. So among these three the LOSO is good and achieved high accuracy. The reason is that LOSO validation is good for small dataset. Therefore, we recommend LOSO validation for this data set as compared hold out and k-folds.

  1. Figure 2 does not make any sense here. If you want to keep it, you should improve it (it is not that informative now, e.g., you can add section number...)

Reply: Thank you. The figure 2 is drawn professionally in updated manuscript.

Reviewer 3 Report

It is good to see that the title of the paper has changed.

Looking at this submitted paper, it is difficult to see what has been changed. I made over 40 comments in the past review, yet the authors seem to have only tried to select 5 comments to address. Overall, the paper is not in a state that I would have liked to have been submitted.

Some issues below:

In the first review I stated "On first impression, it seems the formatting after the title and before abstract is incorrect. Have the authors modified the template? Certainly, needs fixing"
>> This has just not been fixed.

Why has "Tel.: (optional; include country code; if there are multiple corresponding authors, add author initials) +xx-xxxx-xxx-xxxx (F.L.) " not been removed?

Referencing has not been fixed, e.g.:
"[2] and [6]. "

In the main text, the references jump from 1..6 then to 50. What happened to 7..49?

"K. Kayaer and T. Yıldırım, [7] " should be "Kayaer and Yıldırım [7]

"HasanTemurtas et. [8] " should be "Temurtas et al. [8]"

"K. Polat and S. Güneş [9] " should be "Polat and Güneş [9]

"A. M. Sagir and S. Sathasivam [10] " should be "Sagir and Sathasivam [10]"

The paper still has a significant number of poor citation style that just has not been addressed. These errors have not been fixed and are throughout the paper.

The reference list still contains many errors, e.g. reference 1 "Nahla Barakat,Andrew P. Bradley,Mohamed Nabil and H. Barakat, "Intelligible..." should be "N. Barakat, A. P. Bradley, M. Nabil and H. Barakat, "Intelligible...".

"Figure 1 is not of publishable quality and needs to be redone"
>> Is still not of publishable quality

"Table 1 is clearly far too big and is outside of the margins"
>> still too big - why is the font size bigger than the main text?

Figure 2 is still outside of margins!

Figs 3, 4, 5 and 7 are still not of publishable quality.

In the novel contributions section, "1) To design ML based e-health care diagnosis system for diabetes diagnosis. "
>> is NOT new.

The next added to the end of conclusion is incorrect. The highest numbered reference I can see in the main text is "[60]" yet the conclusion introduces new references [64] and [65] without them being discussed in the main section. Conclusion should not introduce new work. Furthermore what happened to references 61..63? In the past review this was raised as an issue and it still has not been fixed.

Overall, my opinion of this paper has not changed. It is very difficult to review the paper as the references seem wrong and they seem not to have been fixed.

Author Response

Reviewer # 3 comments:

it is good to see that the title of the paper has changed.

Looking at this submitted paper, it is difficult to see what has been changed. I made over 40 comments in the past review, yet the authors seem to have only tried to select 5 comments to address. Overall, the paper is not in a state that I would have liked to have been submitted.

Reply: According to previous review many parts, figures and tables have been removed and I incorporated many of your comments.

Comment # 1:

in the first review I stated "On first impression, it seems the formatting after the title and before abstract is incorrect. Have the authors modified the template? Certainly, needs fixing"
>> This has just not been fixed.

Reply: Thank you. We not modified the template and now we used latex template for our manuscript hope all formatting issues have been resolved in updated manuscript.

Comment # 2:

Why has "Tel.: (optional; include country code; if there are multiple corresponding authors, add author initials) +xx-xxxx-xxx-xxxx (F.L.) " not been removed?

Reply: Thank you. Removed in updated manuscript.

Comment # 3:

Referencing has not been fixed, e.g.: 
"[2] and [6]. "

In the main text, the references jump from 1..6 then to 50. What happened to 7..49?

"K. Kayaer and T. Yıldırım, [7] " should be "Kayaer and Yıldırım [7]

"HasanTemurtas et. [8] " should be "Temurtas et al. [8]"

"K. Polat and S. Güneş [9] " should be "Polat and Güneş [9]

"A. M. Sagir and S. Sathasivam [10] " should be "Sagir and Sathasivam [10]".

Reply: Thank you. We updated all references accordingly in updated manuscript.

.

Comment # 4:

The paper still has a significant number of poor citation style that just has not been addressed. These errors have not been fixed and are throughout the paper.

The reference list still contains many errors, e.g. reference 1 "Nahla Barakat,Andrew P. Bradley,Mohamed Nabil and H. Barakat, "Intelligible..." should be "N. Barakat, A. P. Bradley, M. Nabil and H. Barakat, "Intelligible...".

Reply: Thank you.  We incorporated your suggestions regarding references in updated manuscript.

Comment # 5:

Figure 1 is not of publishable quality and needs to be redone"
>> Is still not of publishable quality

"Table 1 is clearly far too big and is outside of the margins"
>> still too big - why is the font size bigger than the main text?

Figure 2 is still outside of margins!

Figs 3, 4, 5 and 7 are still not of publishable quality.

Reply: Thank you. All figures and table 1 have been adjusted in updated manuscript.

Comment # 5:

The next added to the end of conclusion is incorrect. The highest numbered reference I can see in the main text is "[60]" yet the conclusion introduces new references [64] and [65] without them being discussed in the main section. Conclusion should not introduce new work. Furthermore, what happened to references 61.63? In the past review this was raised as an issue and it still has not been fixed.

Overall, my opinion of this paper has not changed. It is very difficult to review the paper as the references seem wrong and they seem not to have been fixed.

Reply: Thank you. All references and conclusion of the paper has been updated accordingly.

Round 2

Reviewer 1 Report

Authors have significantly improved the quality of the presentation and have largely addressed my major concerns.

Please, note that "IOT" is written as "IoT".

You can also improve the quality of Figures 7 & 8.

In the abstract, you said "machine learning data mining" just use either machine learning or data mining!

You also said the performance of your proposal is "high", state specifically how high it is.

Author Response

Reviewer 1: comments:

Authors have significantly improved the quality of the presentation and have largely addressed my major concerns.

Comment # 1

Please, note that "IOT" is written as "IoT".

Reply: Thank you.  We incorporated your comments in updated manuscript.

Comment # 2

In the abstract, you said "machine learning data mining" just use either machine learning or data mining!

Reply: Thank you.  We incorporated your comments in updated manuscript.

Comment # 3

You also said the performance of your proposal is "high", state specifically how high it is.

 Reply: Thank you:  In sections 4.2.7 and 4.2.9 we discussed in details that in term of accuracy the performance of our method is high as compared to other methods.

Reviewer 2 Report

Thanks for the revision.

Author Response

Respected reviewer

Thank you all for valuable and helpful comments

Reviewer 3 Report

This resubmitted paper is much better.

The English needs a lot of work throughout:

1) title - "Recognition of Diabetes in the E-HealthCare", remove "the"

2) "Abstract: A significant attention has been made to the accurate detection", should be "Abstract: Significant attention has been made to the accurate detection"

3) "Diabetes disease (DBD) is a big health issue from which many people are suffered around the world" should be "Diabetes disease (DBD) is a significant health issue that many people suffer from around the world"

4) "To use the classifier DT and the performance have been checked on original features" - I do not know what that really means. Do you mean to use the classifier and verify the performance using the original features?

 etc... I propose the authors have the manuscript checked for English.

The paper states "and can easily be deployed in e-healthcare environment". This assertion is incorrect. It is far from easy to deploy such systems. The paper resents no evidence that it is easy, and the paper has not actually done a deployment to make the assertion.

In the "contributions/novelty" section:

"To design ML based e-health care diagnosis method for diabetes diagnosis." is not novel, others have done this. The authors need to say what is novel about their system.

In the second bullet point - evaluation is not a novel contribution, its validation.

The referencing is much better, but still not right. "et.al" should be "et al.". "In [30]" is presented incorrectly, and "In this study [13] developed" makes no sence. "Ani R et al. [33]" should be "Ani et al. [33]". "Zhe Yang et al. [34]" should be " Yang et al. [34]". "Similarly, in [37] developed" makes no sense, likewise "and information gain [40] and [41]."

on Line 140, "et al." has been italicised, yet not elsewhere.

Figure 1 is STILL a poor-quality bitmap and unsuitable for publication.

Figure 2 is STILL a poor-quality bitmap and unsuitable for publication.

Figure 3 is STILL a poor-quality bitmap and unsuitable for publication.

Figure 4 is STILL a poor-quality bitmap and unsuitable for publication.

Figure 5 is STILL a poor-quality bitmap and unsuitable for publication. Why are the axis fonts twice the size as other fonts?

Figure 6 is STILL a poor-quality bitmap and unsuitable for publication. It cannot be read.

Figure 7 is STILL a poor-quality bitmap and unsuitable for publication. It cannot be read.

It is important to be consistent. Sometimes the paper refers to "E-HealthCare" while sometimes "smart health care system"

In the conclusion, why is "RANDOM FOREST" is all capitals? It is not elsewhere in the paper. 

In the conclusion, why is "ADA-BOOST" being used hyphenated and in all capitals when elsewhere it is "Ada Boost"?

The Author contributions section is missing.

The reference list is better but still incorrect. Journals should have their name capitalised, missing DoI's, reference 63 is incorrect.

Overall, this paper is much better, but to be fair is currently in the state one would expect a first submission to be. Actually, the science seems rather OK and of the same level as other ML papers.

Author Response

Comment # 1:

This resubmitted paper is much better.

Reply: Thank you.

Comment # 2:

title - "Recognition of Diabetes in the E-HealthCare", remove "the"

Reply: Thank you.  The title has been modified accordingly. “Intelligent Machine Learning Approach for Effective Recognition of Diabetes in E-HealthCare  Using Clinical Data”.

Comment # 3:

"Abstract: A significant attention has been made to the accurate detection", should be "Abstract: Significant attention has been made to the accurate detection"

Reply: Thank you. The comment has been incorporate in updated manuscript accordingly. “Significant attention has been made to the accurate detection of diabetes which is a big challenge for the research community to develop a diagnosis system to detect diabetes in a successfully way in the e-healthcare environment”.

Comment # 4:

 "Diabetes disease (DBD) is a big health issue from which many people are suffered around the world" should be "Diabetes disease (DBD) is a significant health issue that many people suffer from around the world"

Reply: Thank you. The sentence has been corrected accordingly in updated manuscript.

Comment # 5:

To use the classifier DT and the performance have been checked on original features" - I do not know what that really means. Do you mean to use the classifier and verify the performance using the original features?

Reply: Thank you.  The Classification performance of classifier has been checked on original features set and on selected features set selected by FS algorithm along with cross validation methods, such as Hold out, K-fold, and LOSO. See Table 6

Comment # 6:

The paper states "and can easily be deployed in e-healthcare environment". This assertion is incorrect. It is far from easy to deploy such systems. The paper resents no evidence that it is easy, and the paper has not actually done a deployment to make the assertion.

Reply: Thank you. Your comment has been incorporated in updated manuscript accordingly.

Comment # 7:

To design ML based e-health care diagnosis method for diabetes diagnosis." is not novel, others have done this. The authors need to say what is novel about their system.

In the second bullet point - evaluation is not a novel contribution, its validation.

 Reply: Thank you. We incorporated your comments in updated manuscript. The first built has been removed. Sure these are validation techniques but we used it in our proposed method.

Comment # 8:

The referencing is much better, but still not right. "et.al" should be "et al.". "In [30]" is presented incorrectly, and "In this study [13] developed" makes no sence. "Ani R et al. [33]" should be "Ani et al. [33]". "Zhe Yang et al. [34]" should be " Yang et al. [34]". "Similarly, in [37] developed" makes no sense, likewise "and information gain [40] and [41]."

on Line 140, "et al." has been italicised, yet not elsewhere.

Reply: The references corrected accordingly.

Comment # 9:

Figure 1 is STILL a poor-quality bitmap and unsuitable for publication.

Figure 2 is STILL a poor-quality bitmap and unsuitable for publication.

Figure 3 is STILL a poor-quality bitmap and unsuitable for publication.

Figure 4 is STILL a poor-quality bitmap and unsuitable for publication.

Figure 5 is STILL a poor-quality bitmap and unsuitable for publication. Why are the axis fonts twice the size as other fonts?

Figure 6 is STILL a poor-quality bitmap and unsuitable for publication. It cannot be read.

Figure 7 is STILL a poor-quality bitmap and unsuitable for publication. It cannot be read.

 Reply: All figures updated corrected accordingly.

Comment # 10:

It is important to be consistent. Sometimes the paper refers to "E-HealthCare" while sometimes "smart health care system"

 Reply:  Thank you: We incorporated your comments accordingly.

Comment # 11:

In the conclusion, why is "RANDOM FOREST" is all capitals? It is not elsewhere in the paper. 

In the conclusion, why is "ADA-BOOST" being used hyphenated and in all capitals when elsewhere it is "Ada Boost"?

Reply:  Thank you: we incorporated your comments and updated conclusion accordingly.

Comment # 12:

The Author contributions section is missing.

Reply:   In section 3 the contribution of the paper presented in details.

Comment # 13:

The reference list is better but still incorrect. Journals should have their name capitalised, missing DoI's, reference 63 is incorrect.

Reply:   we updated the reference accordingly.

This manuscript is a resubmission of an earlier submission. The following is a list of the peer review reports and author responses from that submission.

Round 1

Reviewer 1 Report

This article presents a "new" intelligent approach for effective recognition of diabetes in the IoT-E-Healthcare environment. The authors have used the popular decision tree and ensemble algorithms such as Ada Boost and Random Forest algorithm for feature selection whereas a decision tree classification algorithm has been used to determine whether a patient is healthy of diabetic.

Having reviewed this article, these are my comments.

First of all, authors claimed the system is meant for "effective recognition of diabetes in the IoT E-HealthCare environment". They claim " we have been designed an IOT intelligent decision system based on machine learning algorithms…" This only appears in the title of the article. Nothing about IoT is dealt with in the article. Not even a use case let alone implementation in IoT environment have been addressed.

Secondly, I do not see any innovation or novelty in the article other than using decision tree based algorithms for feature selection. No new data, no new algorithm, no application! What is the benefit of the work proposed, what makes it different from existing ones such as those based on deep learning for automatic feature selection? The authors need to clearly answer these questions!

Third, the article is poorly written with loads of typos and grammatical issues on every page, which makes it painful reading the article. The issues are too numerous to list here. The article needs to be re-written!

In case the article is considered for publication, I will suggest that a separate section be dedicated to related work. Presently the related work has been written as part of the introduction.

Also, figures such as Figure 1 have been poorly drawn with poor resolution. Many figures are not well placed in the text body. Figures on Pages 12 and 13 are not properly labelled. Table 7 has its label in a separate page from where the table is!

Equations (7-12) need to be well formatted.

On page 10, authors define  LOSO as a cross validation method in which samples of the data is split to equal number of samples… I disagree with this definition.

Reviewer 2 Report

The manuscript “A New Intelligent Approach for Effective Recognition of Diabetes in the IoT E-HealthCare Environment” presents a learning-based method to diagnose diabetes. The method relies on standard feature-selection and machine-learning approaches.

The manuscript addresses a relevant problem but presents several weaknesses. The proposed methodology is not innovative and several methodological choices are not justified enough. The authors should make an effort to improve the manuscript readability by (i) shortening some sections, (ii) improving English, (iii) better organizing concepts and figures in the manuscript.

My specific comments can be found hereafter.

Abstract

- Rephrase the sentence at line 23. I suggest writing: “The proposed approach includes three steps: preprocessing, feature selection and classification”

- The data on which the analysis is performed should be clearly reported in the abstract. 

- Lines 26-31 are quite confusing. The need for the “new” filter method should be explained and justified. Which is the difference with respect to the feature selection strategy based on adaboost and random forest? 

- It is not clear if the proposed method is based on decision tree, while random forest and adaboost are only used for comparison. Please, specify.

- All acronyms should be defined (e.g., GL, DPF, and BMI).

Sec. 1

- Please clarify from the beginning if you are dealing with type 1 or type 2 diabetes. Then, focus only on the addressed diabetes type.

- Line 57-68: This paragraph can be heavily summarized. No references are included, thus the paragraph remains qualitative.

- Also in the introduction, please state clearly which are the data you are working on.

- Line 118-127: Considering what the authors propose, this is not a strong discussion on the limits in the literature. For example, the prediction accuracy values of methods in the literature highly depend on the processed dataset: you cannot compare performance obtained with different datasets. Similarly, the computational time depends on the hardware. The authors are proposing a machine-learning algorithm exactly as other researchers do in the literature. So, the innovation with respect to the literature is not clear. 

- I really do not see how the proposed method overcome the literature. I would rather say that the proposed method is another application of machine learning in the diabetes prediction. 

- Line 130-133: There is a lot of confusion here between filtering approaches (which, I imagine, refer to feature selection) and classification approaches (such as AdaBoost and Random Forest). 

- The authors state that: “The proposed system has been tested on the diabetes data set.”. However, no information is given on the diabetes dataset (not even a reference). 

- Figure 1 is useless here, also because we are not dealing with "business", but with people with diabetes. Moreover, the figure is also really blurred.

- The “wireless sensor” only appears at line 148. Please, introduce it properly.

- Line 152: which is the proposed algorithm is still not clear: are you referring to the filtering approach? The decision tree? Random forest..? Please, clarify

- Line 156: training/testing is called hold out.

- I do not agree with considerations done in Table 1, especially for the sentences on computational complexity. The majority of the listed models offer real time performance. Hence, nowadays standard machine-learning algorithms provide real time inference (low computational cost). As for the performance values, these strongly depends on the dataset. 

Sec. 2

- Sec 2.1 should be strongly improved by adding relevant information on the processed dataset.

- Sec. 2.2.1 is non informative and could be summarized/ removed

- Did the authors take inspiration from studies in the literature for designing the feature selection methods? If so, relevant literature should be cited.

- Please, better format all the equations. Otherwise following these subsections of filtering is rather challenging.

- Sec. 2.4 can be removed.

- Table 3 in Sec. 2.5 can be removed.

- Sec. 2.6: the authors should not describe the tests: this is not a book, but a research paper. So the test description is useless.

- As a general comment, figures are blurred. Please, fix.

Sec. 3

- The authors write “The performance of DT classifier was good as compared to other classifiers and therefore, we only report the performance of DT in this paper.”. This is not correct. The authors should report all performance and discuss the performance of each classifier/ method. 

- Results are presented in a confusing way. Images are blurred and tables not well formatted (e.g. Table 5 - Min Max column).

- Which is the reason behind comparing LOSO, k-fold and hold out in Fig. 8? You cannot compare performance on different test set.

- I strongly suggest the authors to shorten this section. There are too many information and the reader lost the point of the discussion.

Sec. 4

- The authors limit the discussion on future work only on the feature selection topic. I suggest to give a wider overview on the learning-based approaches for diabetes treatment. Examples include: [Migliorelli, Lucia, et al. "MyDi application: Towards automatic activity annotation of young patients with Type 1 diabetes." 2019 IEEE 23rd International Symposium on Consumer Technologies (ISCT). IEEE, 2019.]; [Spänig, Sebastian, et al. "The virtual doctor: An interactive clinical-decision-support system based on deep learning for non-invasive prediction of diabetes." Artificial intelligence in medicine 100 (2019): 101706.]

Reviewer 3 Report

On first impression, it seems the formatting after the title and before abstract is incorrect. Have the authors modified the template? Certainly, needs fixing.

The abstract is far too long. "IOT" should be "IoT". " Internet of Things (IOT) has emerging role" - not really, it’s here now! The abstract states "The existing diagnosis systems have some drawbacks, such as high computation time, and low prediction accuracy" then "To handle these issues, we have proposed a IOT based diagnosis system using machine learning methods, such as reprocessing of data, feature selection, and classification for the detection of diabetes disease in e- healthcare environment" but the second sentence does not say anything about addressing the issues in the first sentence. The abstract states "Additionally, the proposed system performance is high as compared to the previous state-of-the-art methods" but the metric for performance has not been given. The abstract states "can easily be deployed in IOT wireless sensor technologies based e-healthcare environment" but this has not at all been proven or demonstrated in the paper.

Poor citation style, "In [7]", "In [8]", etc.

All the references need to be checked, for example "Kemal et al. [9] designed" yet the reference list has "9. K. Polat and S. Güneş". Likewise "Abdu et al. [10] proposed" yet the reference list has "10. A. M. Sagir and S. Sathasivam". It is difficult to seriously assess the paper when the references are all wrong.
In fact the references are a mess as the numbering goes 1-21, 43, 22, 23, 42, 61 .... Obviously Vancouver referencing needs to be in numerical order.

Figure 1 is not of publishable quality and needs to be redone.

In the "contributions/novelty" section:

"I. To design IOT wireless sensor based e-health care diagnosis system for diabetes "detection." is not new and needs to be removed from the contributions/novelty section.

"IV. To recommend that the proposed method can be used to effectively detect the diabetes disease and the system can be easily incorporated in healthcare. The performance of the proposed method in terms of accuracy is high as compared to other states of the art methods and we analyzed it statistically" - this is aim not novel contribution and needs to be removed from the contributions/novelty section.

Table 1 is clearly far too big and is outside of the margins.

Figure 2 is not of publishable quality and needs to be redone.

The text formatting in section 2.2.1 seems incorrect.

There are spelling mistakes in algorithm 1.

There are spelling mistakes in algorithm 2.

Table 4 is clearly far too big and is outside of the margins.

Figure 3 is far too big, not of publishable quality and needs to be redone.

Figure 4 is far too big, not of publishable quality and needs to be redone.

Table 5 is clearly far too big and is outside of the margins.

Figure 5 is far too big, not of publishable quality and needs to be redone.

Figure 6 and 7 seem to have huge axis text.

Table 9 is outside of the margins.

Figure 8 is crazy big.

Figure 9 is poor quality to be read

Table 11 is obviously outside of the margins.

Figures 10, 11, 14 and 15 are crazy big.

Table 12 is obviously outside of the margins.

The reference list is poorly presented and all needs to be redone.

There is a clear and obvious lack of attention to the paper.

I cannot find which dataset has been used for the research.

After reading the paper it is so poorly presented, I am having difficulty separating this from the presented research. Taking a step back, the paper says it’s to do with IoT yet there is no IoT issues in this paper at all. It can clearly be implemented outside of IoT systems. The paper is fundamentally a decision tree paper and should be presented as such.